



# SAGE III/ISS Ozone and NO₂ Validation using Diurnal Scaling Factors

Sarah A. Strode[1,2], Ghassan Taha[1,2], Luke D. Oman[2], Robert Damadeo[3], David Flittner[3], Mark Schoeberl[4], Christopher E. Sioris[5], Ryan Stauffer[2]

[1]Morgan State University, Baltimore, MD, USA 21251
[2]NASA Goddard Space Flight Center, Greenbelt, MD 20771
[3]NASA Langley Research Center, Hampton, VA, USA
[4]Science and Technology Corporation, Columbia, MD, USA
[5]Air Quality Research Division, Environment and Climate Change Canada, Toronto, Canada

*Correspondence to*: Sarah A. Strode (sarah.a.strode@nasa.gov)

**Abstract.** We developed a set of solar zenith angle, latitude- and altitude-dependent scaling factors to account for the diurnal variability in ozone and NO₂ when comparing Stratospheric Aerosol and Gas Experiment (SAGE) III/ISS observations to observations from other times of day. The scaling factors are calculated as a function of solar zenith angle from the 4-dimensional output of a global atmospheric chemistry model simulation of 2017-2020 that shows good agreement with observed vertical profiles. Using a global atmospheric chemistry model allows us to account for both chemically and dynamically driven variability. Both year-specific scale factors and a multi-year monthly climatology are available to decrease the uncertainty in inter-instrument comparisons and allow consistent comparisons between observations from different times of day. We describe the variability in the shape of the diurnal scale factors as a function of space and time. The quasi-biennial oscillation (QBO) appears to be a contributing factor to interannual variability in the NO₂ scaling factors, leading to differences between years that switch sign with altitude. We show that application of these scaling factors improves the comparison between SAGE III/ISS and OSIRIS NO₂, and between SAGE III/ISS and OMPS LP, OSIRIS and ACE-FTS ozone observations. The comparisons between SAGE III/ISS ozone for sunrise or sunset versus MLS daytime or nighttime observations are also more consistent when we apply the diurnal scaling factors. There is good agreement between SAGE III/ISS V5.2 ozone and correlative measurements, with differences within 5% between 20-50 km when corrected for diurnal variability. Similarly, the SAGE III/ISS V5.2 NO₂ agreement with correlative measurement is mostly within 10%. While the scale factors were designed for use with SAGE III/ISS observations, they can easily be applied to other observation intercomparisons as well.

## 1 Introduction

Observations from the Stratospheric Aerosol and Gas Experiment (SAGE) III began in 2017 following its successful docking with the International Space Station (ISS). SAGE III/ISS measures vertical profiles of ozone (O₃), nitrogen dioxide (NO₂), and water vapor, as well as cloud presence using solar occultation measurements (McCormick et al., 1989; Wang et al., 2006; Schoeberl et al., 2021). Observations are thus available at both sunrise and sunset. It also provides profiles of aerosol extinction at multiple visible, near infrared and ultraviolet wavelengths. SAGE III/ISS



extends the SAGE series of solar occultation instruments that began with the Stratospheric Aerosol Measurement (SAM) in July 1975 and includes SAM II, which flew from 1978-1993; SAGE I, which launched in 1979; SAGE II, launched in 1984; and SAGE III Meteor, launched in 2001. SAGE I/II instruments were heavily used in long-term trend studies because of their precise measurements and long data record (WMO, 1988, 2011; Harris et al., 2015). Accurate, continuous measurements of stratospheric $NO_2$ are necessary because of the important role of $NO_2$ in the
Earth's ozone distribution (Crutzen, 1979).

     Stratospheric $NO_2$ experiences a strong diurnal cycle. Photolysis of $NO_2$ leads to a rapid drop in concentration at sunrise, while $NO_2$ concentrations rapidly rise at sunset as NO is converted to $NO_2$ (e.g. Brohede et al., 2007; Solomon et al., 1986 and refs therein). Ozone also experiences a diurnal cycle due to photochemistry. This cycle is large in
the upper stratosphere and mesosphere (e.g. Vaughan, 1982; Prather, 1981), but also exceeds 2% percent in the middle stratosphere (Sakazaki et al., 2013; Parrish et al., 2014). Model simulations suggest diurnal variability in the tropospheric ozone column can reach over 9 DU in some locations and changes over time due to evolving precursor emissions (Strode et al., 2019). Damadeo et al. (2018) found that biases in diurnal sampling in occultation instruments can affect ozone trend calculations. Accounting for the diurnal cycle above 35 km allows a more direct comparison
between SAGE III/ISS observations and observations from instruments that measure at different times of day, such as the Microwave Limb Sounder (MLS) (Waters et al., 2006) on the Aura satellite (Schoeberl et al., 2006), which measures ozone at mid-day and in the middle of the night outside of the polar regions, where sampling occurs over a wider range of local times. Estimates of the diurnal variability also provide a basis for comparison of the sunrise versus sunset measurements with SAGE III/ISS (Wang et al., 2020).


     Previous studies often used the PRATMO (Prather, 1992; Prather and Jaffe, 1990) photochemical box model to account for diurnal variability in $NO_2$ when comparing observations from different times of day (Brohede et al., 2007; Dubé et al., 2020) and to account for $NO_2$ variability along the line of site (Dubé et al., 2021). In order to account for differences in sampling times between ozone instruments, Frith et al. (2020) used a global model simulation to develop
a climatology of ozone diurnal variability based on time of day.

     In this work, we create diurnal scaling factors for ozone and $NO_2$ as a function of solar zenith angle (SZA), latitude, and altitude for each month and year of the SAGE III/ISS period. We use a global model to account for vertical, horizontal, and temporal differences in $NO_2$ and $O_3$ due to both chemistry and transport. Studer et al. (2014) found
interannual variability in the diurnal cycle of stratospheric and mesospheric ozone above Switzerland. We therefore develop year-specific diurnal scale factors as well as climatological diurnal scaling factors. The resulting scale factors are publicly available and provide a convenient resource for accounting for the diurnal cycle when comparing observations from SAGE III/ISS or other instruments to observations from other times of day. This allows a greater number of observations to be directly compared since the observations can occur at different times of day.




We describe the model and methods used to develop diurnal scaling factors in Section 2 and evaluate the simulated $O_3$ and $NO_2$ with observations in Section 3. Section 4 presents the geographic and temporal variability of the scaling factors and demonstrates their application to measurement comparisons for $NO_2$ and $O_3$. We present conclusions in Section 5.

## 2 Methods

### 2.1 Instrument Descriptions

#### 2.1.1 SAGE III/ISS

The SAGE III/ISS instrument was launched to the International Space Station (ISS) on February 19, 2017. The instrument scans over the sun during sunrise and sunset events, measuring the atmospheric extinction along the line of sight (Cisewski et al., 2014). SAGE III/ISS profiles are produced on a 0.5 km grid with an estimated vertical resolution of 0.7 km from 10-50 km for $NO_2$ and 6-85 km for $O_3$ (SAGE III Algorithm Theoretical Basis Document, 2002). SAGE III coverage and number of profiles is limited to about 15 sunrise and 15 sunset events per day, with the majority of observations occurring between 60°S and 60°N. We use the "aerosol ozone" (AO3) ozone retrieval, which is similar to the SAGE II retrieval method (Damadeo et al., 2013), as recommended by Wang et al. (2020). Wang et al (2020) reported that the V5.1 ozone profile has 5% accuracy between 15-55 km and 3% precision between 20-40 km. They also reported a 5-8% sunrise/sunset bias in the upper stratosphere that they could not explain. However, the Wang et al. (2020) analysis did not account for ozone diurnal variability and attributed the larger bias above 45 km to the ozone diurnal cycle. Dubé et al. (2021) reported that the SAGE III/ISS $NO_2$ V5.1 is 20% biased high in the mid-stratosphere when accounting for diurnal variability. The difference between V5.2 and V5.1 ozone is less than 0.5% and resulted from various algorithm improvements, while the $NO_2$ in V5.2 decreased by 5%, which was caused mainly by the new wavelength map (SAGE III/ISS V5.2 release notes, 2021). Additional changes include better oxygen dimer ($O_4$) corrections and the removal of all vertical smoothing.

#### 2.1.2 OSIRIS

The Optical Spectrograph and InfraRed Imaging System (OSIRIS) instrument (Llewellyn et al., 2004) is a limb sounder that was launched on February 2001 onboard the Odin satellite (Murtagh et al., 2002). OSIRIS provides vertical profiles of ozone, aerosol and $NO_2$ with approximately 2 km vertical resolution. Variations in SZA along the line of sight can impact retrievals of species with strong diurnal cycles such as $NO_2$ for occultation and limb measurements (Mclinden et al., 2006; Brohede et al., 2007). The reported accuracy of the OSIRIS V6.1 $NO_2$ retrieval is ±10% when accounting for the diurnal variability in $NO_2$ along the line of sight (Sioris et al., 2017), and 5% above 21 km for the ozone v5.07 retrieval (Adams et al., 2014).



### 2.1.3 MLS

The Microwave Limb Sounder (MLS) (Waters et al., 2006) was launched on the Aura satellite (Schoeberl et al., 2006) in July 2004 and provides global observations of trace gases including ozone. MLS ozone observations extend from
the upper troposphere to the mesosphere. We use MLS V4.2 $O_3$ observations, since the differences in stratospheric $O_3$ compared to version 5 are small (Livesey et al, 2022). We use MLS data from both early afternoon and night-time overpasses. The accuracy of MLS $O_3$ measurements varies with altitude, ranging between 5-10% from 68-0.2 hPa (Livesey et al., 2020).

### 2.1.4 ACE-FTS

The Atmospheric Chemistry Experiment Fourier Transform Spectrometer (ACE-FTS) (Bernath et al., 2005; Bernath, 2017) measures trace gas profiles from the SCISAT-1 satellite. ACE-FTS, like SAGE III/ISS, uses solar occultation to take measurements during sunrise and sunset. Consequently, comparisons between the V3.6 ACE-FTS and SAGE III/ISS observations do not require correction for the diurnal cycle as long as sunset is compared with sunset and sunrise with sunrise. The ACE-FTS ozone profile accuracy is within 5% between 20-45 km and exhibits a large bias
of 10-20% above 45 km (Sheese et al., 2017). The $NO_2$ accuracy is 20% between 20-40 km (Kerzenmacher et al., 2008).

### 2.1.5 OMPS LP

The Ozone Mapping and Profiler Suite (OMPS) consists of three instruments designed to measure the ozone layer. OMPS is on board the Suomi National Polar-orbiting Partnership (NPP) satellite (Flynn et al., 2006), which launched
in October of 2011. The limb profiler (LP) instrument is designed to provide high-vertical-resolution ozone and aerosol profiles from measurements of the scattered solar radiation in the 290–1000 nm spectral range and can provide daily global measurements of ozone and aerosol profiles from the cloud top up to 60 and 40 km, respectively. The V5.2 ozone profiles' accuracy is within 10% at altitude range 18-42 km, except for the northern high latitudes, which has a larger negative bias between 20-32 km, and above 43 km (Kramarova et al., 2018).


### 2.2 GEOS Model Simulation

We use the global three-dimensional Goddard Earth Observing System (GEOS) model (Molod et al., 2015) coupled with the Global Modeling Initiative (GMI) stratospheric and tropospheric chemistry mechanism (Nielsen et al., 2017; Duncan et al., 2007; Strahan et al., 2007) and the Goddard Chemistry Aerosol Radiation and Transport (GOCART)
aerosol module (Chin et al., 2002; Colarco et al., 2010) to simulate the distribution and variability of $O_3$, $NO_2$, and other trace gases and aerosols. GMI uses an updated version of Fast-JX (Bian and Prather, 2002) to simulate photolysis. The GOCART aerosols are coupled to the GMI chemistry and impact the photolysis rates as well as the surface area density (SAD) of polar stratospheric clouds for heterogeneous chemistry. A replay method described by Orbe et al. (2017) is used to constrain the model's meteorology to the MERRA-2 reanalysis (Gelaro et al., 2017). We
refer to this simulation setup hereafter as GEOS-GMI.



The simulation has 72 vertical levels from the surface to 1 Pa, a horizontal resolution of approximately 100 km, and a chemistry time step of 15 minutes. Three-dimensional $O_3$ and $NO_2$ concentrations are output every half hour in order to better resolve the diurnal cycle. We simulate the period from January 2017 through December 2020. In
addition to trace gas concentrations, the model simulation includes several other diagnostics used in this analysis. These include solar zenith angle (SZA), and the tendency of ozone due to chemistry and the tendency due to dynamics. These tendencies quantify the change in ozone in a given grid box caused by local chemical processes versus large-scale transport, and are diagnosed from the change over a given operator in the model.

**2.3 Scaling Factor Calculation**

We construct diurnal scaling factors from the GEOS-GMI model output by taking the ratio of the $O_3$ and $NO_2$ concentrations at each zenith angle to the concentration at sunrise and sunset. For convenience, we use "signed SZA", with negative values for afternoon and positive for morning. We thus define sunrise as SZA = 90° and sunset as SZA = -90°. We interpolate the model output at each latitude/longitude to the SAGE III/ISS geometric altitude levels,
which have a grid spacing of 0.5 km.

While model output is available for every day, we use monthly zonal mean values to construct the scaling factors for each latitude, altitude, and SZA. The diurnal variability of $O_3$ is influenced by dynamics as well as chemistry. Sakazaki et al. (2013) and Sakazaki et al. (2015) highlight the contribution of tidal winds to the diurnal variability of
stratospheric $O_3$. Schanz et al. (2021) report variability in the $O_3$ diurnal cycle due to dynamics in reanalysis fields. We aim to capture the chemistry effects as well as systematic dynamical effects on the diurnal cycle, while filtering out the short-term temporal and spatial variability caused by day-to-day variations in transport. Using monthly and zonal means filters out much of this random variability to create a more reliable picture of the diurnal cycle and the relative role of chemical versus dynamical effects. Examination of the dynamical versus chemical tendencies from
the simulations shows that the diurnal cycle in the $O_3$ tendency from dynamics is important between 40 and 50 km, even in the monthly zonal mean. Figure 1 compares the amplitude of the diurnal cycle, defined here as the maximum of the diurnal cycle minus the minimum, for the chemical and dynamical tendencies of $O_3$ and $NO_2$ for January 2019. While the chemical tendency of $O_3$ is dominant throughout much of the atmosphere above 30 km, the diurnal amplitude of the dynamical tendency term can equal or exceed the amplitude of the chemical term near 45 km in the
tropics. Our calculated scaling factors thus include both chemical and dynamical effects on the diurnal cycle. For $NO_2$, the chemical tendency is dominant throughout the profile. We note that if the tendencies are normalized by the concentration of the constituent, the chemical tendency of $NO_2$ (in % $s^{-1}$) increases with altitude above 45km rather than peaking at 40-50km.

We calculate scaling factors referenced to sunrise and sunset for easy application to SAGE III/ISS data when comparing to observations from different times of day. The factors are provided on an SZA by altitude grid with one



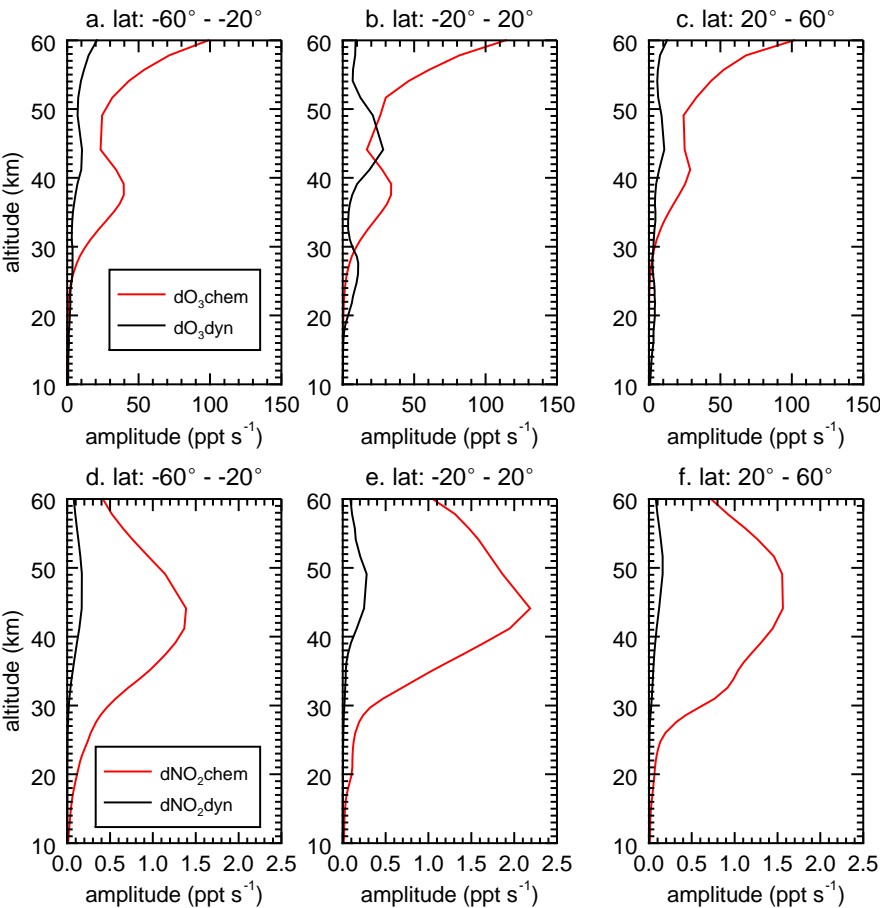

**Figure 1: The amplitude of the diurnal cycle of the simulated ozone tendency (top) and NO₂ tendency (bottom) due to dynamics (black) and chemistry (red) for January 2019, averaged over three latitude bands.**

file per month for January 2017 through December 2020. The SZA grid is nonlinear to allow finer resolution near the terminator, when the values are changing rapidly. In addition to the year-specific scaling factors, we provide a monthly climatology of scaling factors, based on the average of 2017 through 2020, that can be applied to other time periods. We also provide the zonal mean concentrations of $O_3$ and $NO_2$ as function of SZA, latitude, and altitude, so that users can derive their own scaling factors for arbitrary SZA pairs.

## 3 Model Validation

We compare the simulated $NO_2$ and $O_3$ profiles to observations from SAGE III/ISS and other instruments to determine the suitability and limitations of the simulated values for deriving scaling factors.





### 3.1 Comparison to NO₂ Observations

We compare the NO$_2$ from our model simulation to sunrise and sunset observations from SAGE III/ISS. We note that
the SZA diagnosed by the simulation sometimes deviates from that of the SAGE III/ISS observations at the same
location, which by definition is ±90° (depending on sunrise or sunset) and is reported for each event at the average
longitude/latitude/time of all scans through a particular altitude. A mismatch in SZA can lead to disagreement between
the simulated and observed NO$_2$. Consequently, we sample the model by first determining the grid box corresponding
to the SAGE III/ISS observation, and then finding the grid box that best matches the SAGE III/ISS SZA (±90°) at the
observation latitude within 8 grid boxes (approximately 800 km) longitudinally of the observation location. This
sampling methodology improves the agreement between the simulated and observed NO$_2$.

Figure 2 shows the vertical profiles of simulated NO$_2$ compared to SAGE III/ISS observations for sunrise and sunset,
for December through February of 2017-2020. Overall, the model simulation reproduces the major features of the
vertical distribution and latitudinal variations of the SAGE III/ISS observations. The mean values are in good
agreement at many altitudes and latitudes, but the simulation underestimates the SAGE III/ISS sunrise observations
in the troposphere. Dubé et al (2021) found that SAGE III/ISS NO$_2$ is biased high, particularly at lower altitudes. The

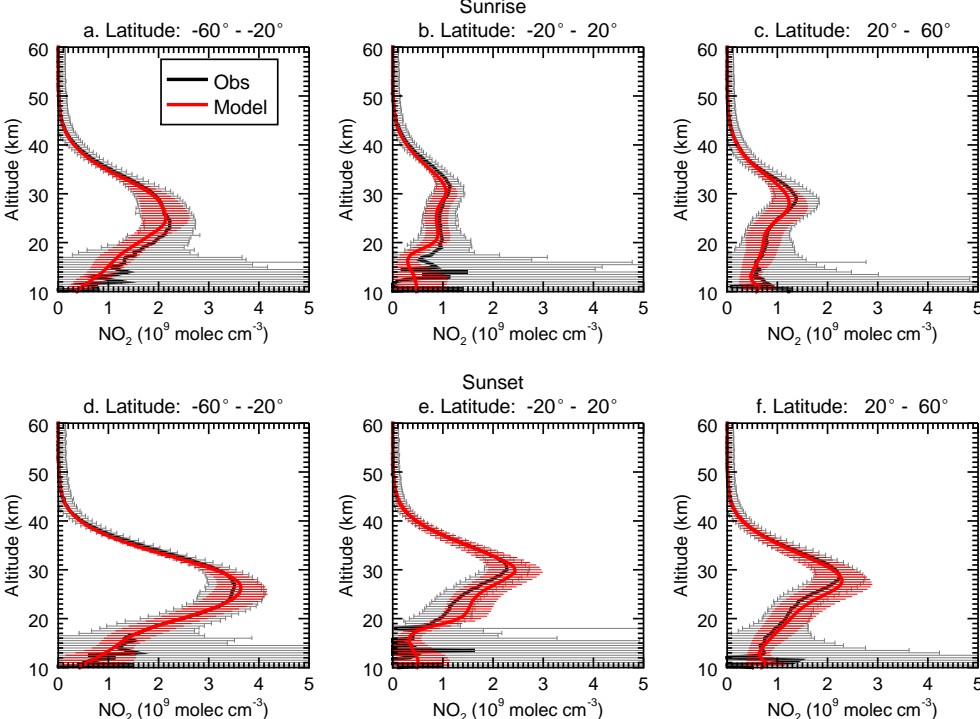

**Fig. 2: Comparison of the model simulation (red) to SAGE III/ISS (black) sunrise (top) and sunset (bottom) NO₂ vertical**
**profile observations for Dec.-Jan.-Feb. of 2017-2020 averaged over three different latitude bands. Error bars represent the**
**standard deviation within the latitude band.**



sunset comparison shows a model overestimate at 20-30 km in the tropics. Between 20 and 40km, the simulated profiles agree with the observed values within 20%, except for the sunset profiles of the 20°S-20°N band, where the model overestimate reaches 40% at 20.5 km. However, comparison of the sunrise and sunset profiles suggests that the simulation is able to capture many of the observed sunrise-sunset differences. Figure S1 shows the sunrise and sunset $NO_2$ comparisons for June-August of 2017-2020. There is good overall agreement between the simulated and observed $NO_2$ in terms of the mean values and the profile shapes, as well as how the profiles change between sunrise and sunset. The simulation underestimates the SAGE III/ISS peak around 30 km and places it slightly too low in the Southern hemisphere. Both the simulation and the observations show lower values around 30 km for sunrise compared to sunset, consistent with the box model results of Dubé et al (2020), since $NO_x$ concentrations increase over the day due to photolysis of $N_2O_5$ and other reservoir species (Belmonte Rivas et al., 2014). Increases in the $NO_2$ column over the day are also seen in FTIR observations (Sussmann et al., 2005).

We also compare the simulated $NO_2$ profiles to observations from the Optical Spectrograph and InfraRed Imaging System (OSIRIS) instrument (Llewellyn et al., 2004; Sioris et al., 2017). Figure S2 shows the comparison for July and August of 2017-2018. The simulation is biased high compared to OSIRIS throughout much of the profile between 10 and 40 km. The low biases seen in the SAGE III/ISS comparison (Fig. S1) are not present in the OSIRIS comparison. Some of this discrepancy may be due to the diurnal differences in $NO_2$ along the line of sight (LOS) (Brohede et al., 2007; Dubé et al, 2021) that are not accounted for in the SAGE III/ISS retrieval.

### 3.2 Comparison to $O_3$ Observations

Previous studies have evaluated the stratospheric ozone and its variability in the GEOS model with GMI chemistry. Parrish et al. (2014) found reasonable agreement between the simulated $O_3$ diurnal cycle at Mauna Loa, Hawaii with microwave ozone profiling radiometer (MWR) observations at most levels, although the diurnal peak relative to midnight is overestimated in the model compared to the MWR observations for 35-39 km. Frith et al. (2020) compared the climatological diurnal $O_3$ cycle from a similar model simulation to the one in this paper to observations from the Superconducting Submillimeter-Wave Limb Emission Sounder (SMILES) and the Upper Atmosphere Research Satellite (UARS) MLS, with good agreement. They also compared the simulated day vs. night $O_3$ differences to Aura MLS observations and the sunrise vs. sunset differences to SAGE III/ISS observations. They found good overall agreement with the structure of the MLS differences, while the simulated sunrise/sunset ratio differed from that of SAGE III/ISS above approximately 45 km.

We present additional validation of the simulated $O_3$ with comparisons to SAGE III/ISS observations and ozonesondes. Figure 3 compares the simulated $O_3$ with SAGE III/ISS observations from Dec.-Jan.-Feb. of 2017-2020 for sunrise and sunset. There is good agreement between the model and observations above approximately 15 km. The model tends to underestimate the observations below 15 km, although the observations show large variability. Between 20 and 50 km, the model profiles for all three bands are within 15% of the observations. The model underestimates the peak $O_3$ between approximately 25-30 km for the 20°S-20°N range. Similar features are seen in



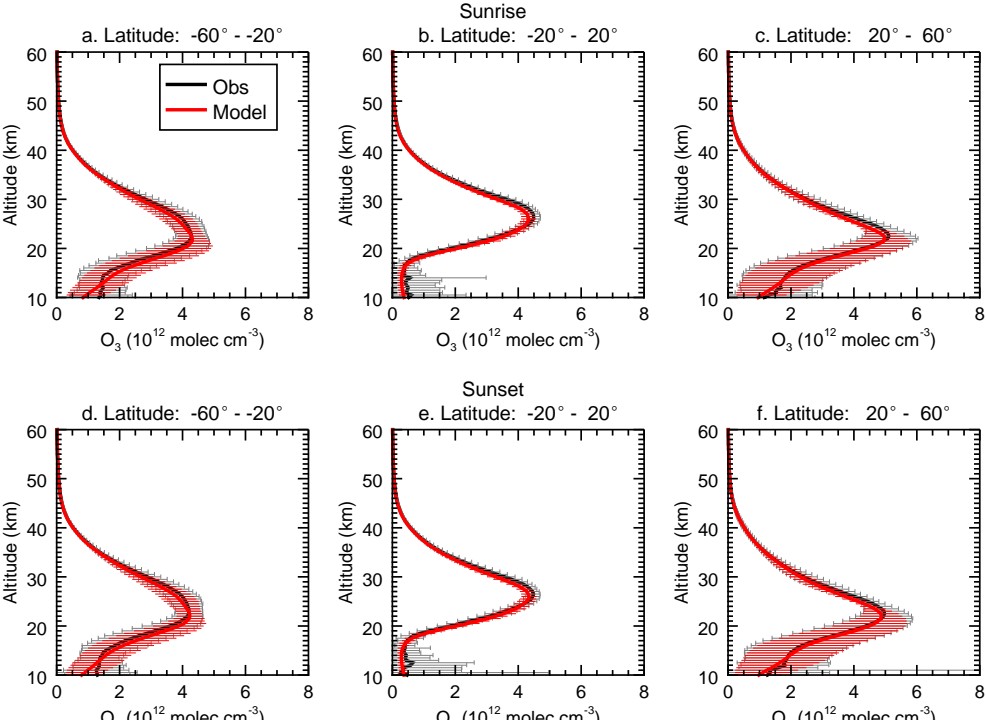

**Fig. 3: Comparison of the model simulation (red) to SAGE III/ISS (black) sunrise (top) and sunset (bottom) O₃ vertical profile observations for Dec.-Jan.-Feb. of 2017-2020 averaged over three different latitude bands. Error bars represent the standard deviation within the latitude band.**

the June-Aug. comparison (not shown), along with a small model overestimate around 15-20 km. Figure S3 shows a comparison of simulated $O_3$ to ozonesonde profiles in three latitude ranges. There is good agreement in the profile shapes and latitudinal differences, but the simulated $O_3$ is biased high in the 15-20 km range. Stauffer et al. (2019) also found a high bias in this region and attributed it partly to the model's limited vertical resolution causing discrepancies in the altitude of the tropopause gradient compared to sondes. The high bias below 10 km seen in the SAGE III/ISS comparison is not present in the ozonesonde comparison.

## 4 Results

### 4.1 Diurnal Scaling Factors for NO₂

In this section we describe the overall shape of the diurnal scaling factors for $NO_2$ as well as their geographic and temporal variability. We then illustrate how application of the diurnal scale factors improves the agreement between observations taken at different times of day.



### 4.1.1 Description of NO₂ Diurnal Scale Factors

We present the climatological scale factors as a function of latitude, altitude, SZA, and month. Figure 4 shows the
climatological sunrise and sunset diurnal scale factors for $NO_2$ as a function of signed solar zenith angle for January
and July at 45°N at 35 km. The U-shape of the scaling factors reflects the high $NO_2$ values at night and low values
during the day, with sharp gradients occurring at sunrise (SZA=90°) and sunset (SZA=-90°). The sunrise and sunset
factors have a similar shape, but are offset in magnitude because the sunrise and sunset values of $NO_2$ differ as
described in section 3.1. Gaps in the plot represent SZA values that do not occur in the monthly mean. A larger gap

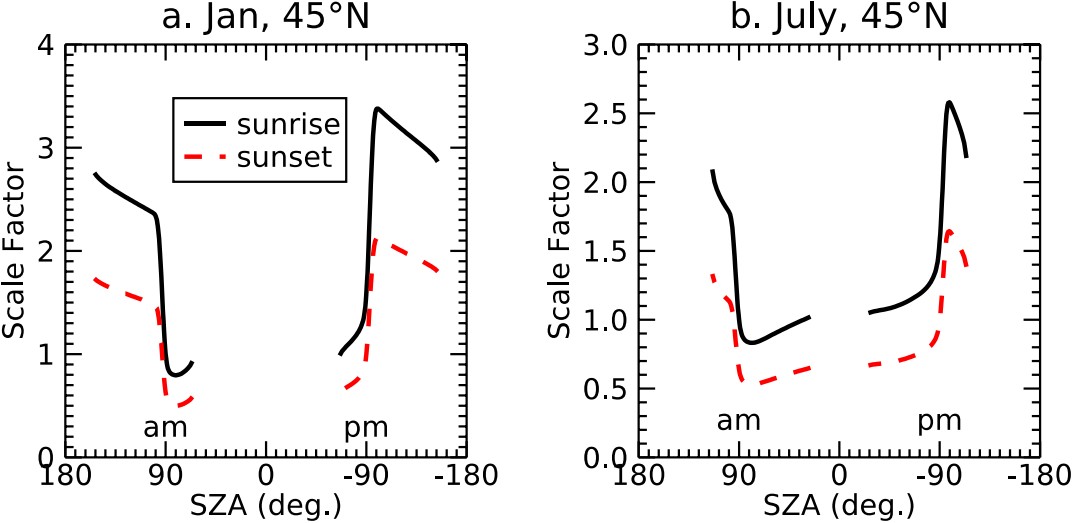

**Fig. 4: Diurnal scaling factors for sunrise (black) and sunset (red) as a function of SZA at 45°N for a. January and b. July
at 35 km altitude. The scaling factors represent the ratio of the $NO_2$ at the given SZA to the values at sunrise or sunset.**

around SZA=0° occurs in January compared to July at 45°N, reflecting the lower sun angle in January. The January
scaling factors also reach a larger maximum value at night compared to the July factors at this latitude. While the
overall shape of the $NO_2$ scaling factors is similar across the altitude range of the SAGE III/ISS measurements, the
magnitude changes dramatically with altitude because of the larger diurnal cycle of $NO_2$ at higher altitudes. Figure
S4 uses a nonlinear color scale to show the large amplitude of the diurnal scaling factors at high altitudes.

We next explore the latitudinal variability in scaling factors, using the sunrise factor for SZA=60° at 35 km altitude
as an example. We show the variations in the scale factor as a function of latitude for one month in each season in
Fig. 5. There is considerable variability in the factor with both latitude and month. January shows the greatest
variability, with values ranging from 0.65 at 69°S to 0.95 at 39°N. Both January and October show the largest
deviation from 1 at the southern end of the range for which SZA=60° is reached, while April and July deviate most
strongly from 1 at the northern end.

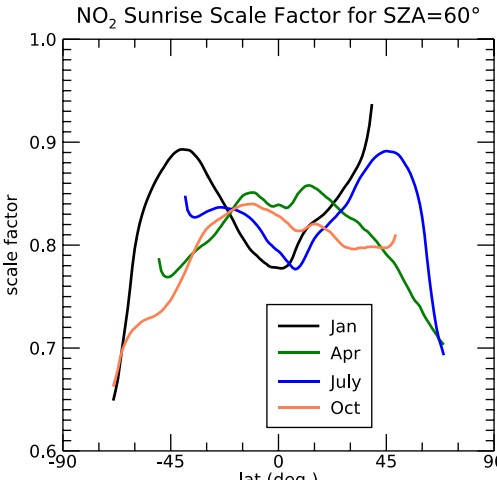

275

**Fig. 5: The NO₂ sunrise scale factor at 35 km for SZA=60 as a function of latitude for January (black), April (green), July (blue), and October (orange).**

### 4.1.2 Interannual variability (IAV) of NO₂ Diurnal Scale Factors

Since we have created diurnal scale factors from both monthly climatological averages and from individual years, we
280    investigate how much IAV exists in the NO₂ diurnal scale factors. Figure 6 shows the IAV in the sunrise NO₂ scaling
factors for October. All four years show a similar shape for the factors as a function of signed SZA at the equator at
25 km (Fig. 6a), but in 2018 the scale factors are larger than the climatology for SZA < 90°, while for 2017 and 2019
they are smaller. The situation is reversed in the southern high latitudes, where 2018 and 2020 are smaller than the
climatology and 2017 and 2019 are greater (Fig. 6b). Figure 6b shows that the percent difference between the
individual years and the climatology is largest near the equator and south of 60°S in October. Considering the
difference from climatology for the SZA=60° factor as a function of altitude, we find that, at the equator, the
differences are largest from approximately 15-35 km, but deviations from climatology do not exceed 15% below 50
km (Fig. 6c). Park et al. (2017) found that the QBO plays a dominant role in the IAV of tropical stratospheric NO$_x$
seen in OSIRIS observations. Zawodny and McCormick (1991) found that QBO variability of SAGE II NO₂ was
related to changes in the vertical transport of NO$_y$ and noted that the time of day could affect the relationship of NO₂
to the QBO. We find that the yearly anomalies in the NO₂ scale factors for the lower stratosphere show a similar
vertical structure to the anomalies in the vertical gradient of the zonal wind anomalies at the equator (Fig. 6f),
indicating that variability associated with the QBO is likely responsible for the interannual variability at these altitudes.

At 60°S, the variability is larger throughout much of the atmosphere and reaches values above 20% near 10-20 km
(Fig. 6d). Considering all latitudes and altitudes below 50 km, the maximum difference between an individual year
and climatology for the SZA=60° factors is 54% in October. The largest difference for the SZA=60° factors when all
months are considered is 75%, which occurs in September at 23.5 km. When all SZA values between -90° and 90°

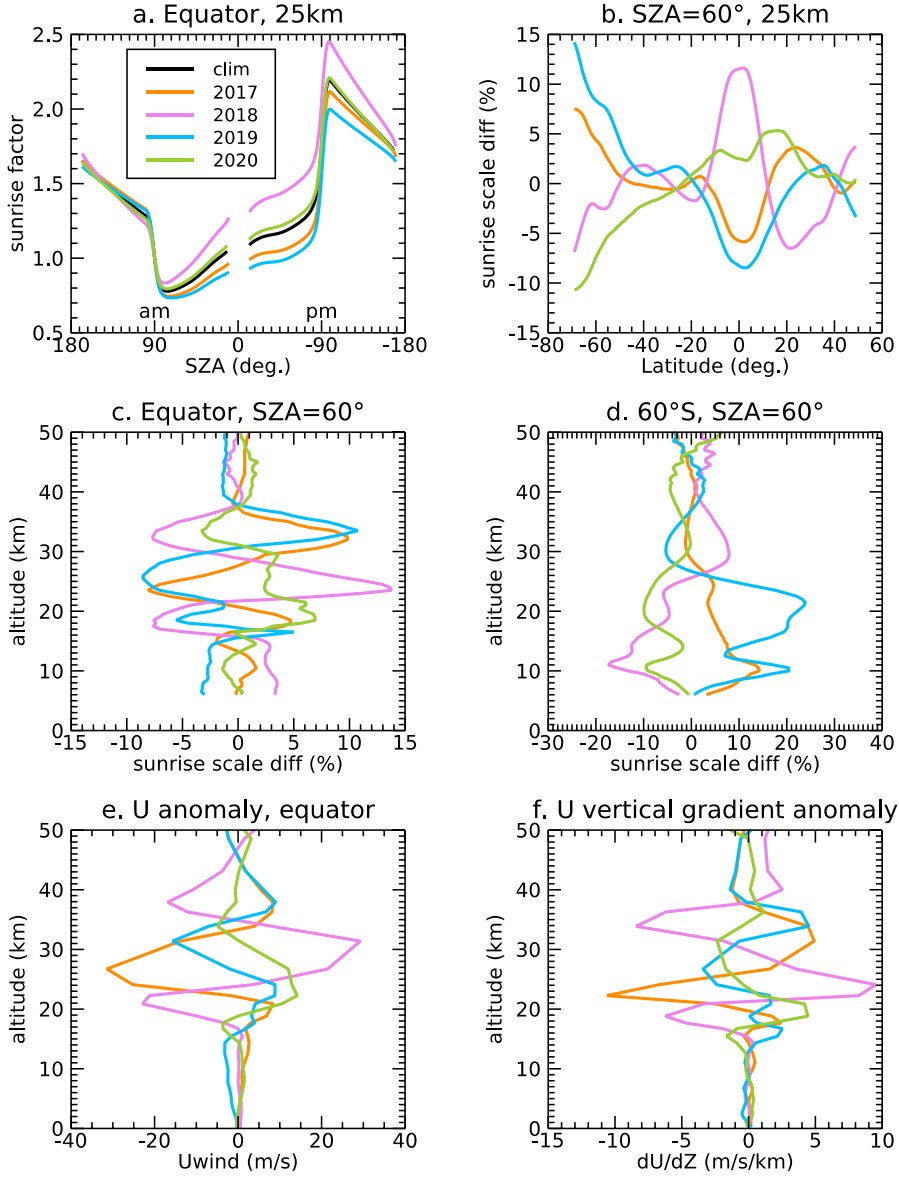

**Fig. 6: Interannual variability in the October sunrise NO$_2$ scaling factors, which are referenced to SZA=90. a. scaling factors as a function of signed SZA for the equator at 25 km for the climatology (black), 2017 (orange), 2018 (magenta), and 2019 (cyan), and 2020 (green). b. percent difference from climatology in the scaling factors for SZA=60 as a function of latitude for each year. c. percent difference from climatology for the SZA=60 scale factors for each year as a function of altitude at the equator. d. percent difference from climatology for the SZA=60 scale factors for each year as a function of altitude at 60°S. e. simulated zonal mean zonal wind speed at the equator as a function of altitude f. the vertical gradient in the zonal wind speed.**





are considered, the maximum difference reaches 118% at 13.5 km in September. However, the IAV differs according
to the month and latitude considered, so many of the differences average out when an entire year or large latitude
range is considered.

### 4.1.3 Application of NO₂ Diurnal Scale Factors

We demonstrate the utility of the NO₂ diurnal scaling factors by comparing SAGE III/ISS NO₂ observations with
observations from OSIRIS with and without the application of the diurnal scaling factors. We also include the solar
occultation ACE-FTS as a reference since it does not require any diurnal corrections when comparing with SAGE
III/ISS. We note that the scale factors are intended to account for the temporal change in concentration between
different observation times, and not to alter the value of the SAGE III/ISS retrieval itself.

Figure 7 shows the percent difference between SAGE III/ISS sunrise (SR) and sunset (SS) NO₂ and OSIRIS and ACE-
FTS observations averaged over three latitude bands before and after applying the diurnal scale factors. The
coincidence criteria between SAGE III and the reference instrument are defined as same-day measurements that are
within 3° latitude and 10° longitude. For ACE-FTS, we matched SR/SS that met the criteria and were within 3 hours
of each other. In general, the disagreement between SAGE III and ACE-FTS for both sunrise and sunset measurements
(magenta and green lines in Fig. 7) is 20% or less for most altitudes. The difference between SAGE III and OSIRIS
(red and purple lines) is large. The difference for sunrise observations exceeds 50% below 20 km and exceeds 25%
below 35 km north of 20°S. Differences are especially large in the tropics below 22 km. Sunset differences exceed
50% throughout much of the atmosphere below 35 km. NO₂ diurnal variability and the mismatch of the measurement

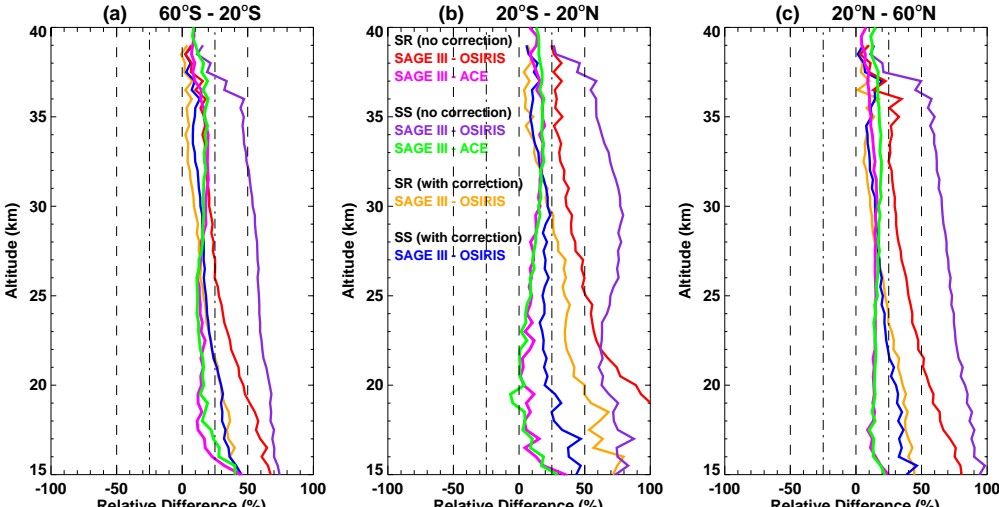

**Fig. 7: The percent difference between SAGE III/ISS sunrise (SR) and sunset (SS) NO₂ and OSIRIS and ACE-FTS**
**observations averaged over three latitude bands. The OSIRIS comparisons without application of diurnal corrections are**
**show in red and purple for sunrise and sunset, respectively, while the comparisons with the diurnal scaling factors applied**
**are shown in yellow and blue. The comparisons to ACE are shown in magenta for sunrise and green for sunset.**





times explains much of these differences. The difference between the two instruments is significantly reduced when accounting for the $NO_2$ diurnal cycle (yellow and blue lines). The difference becomes mostly less than 50% for both
sunrise and sunset and below 25% above 25 km, except for the sunrise observations between 20°S and 20°N. Applying the scaling factors improves the agreement between the SAGE and OSIRIS profiles in all latitude bands (Fig. 7) and improves the consistency between the sunrise and sunset comparisons, particularly in the 20°-60° N and S ranges. The larger difference below 25 km is mostly caused by the diurnal effect error which occurs due to the variation of the SZA along the line of sight in occultation measurement. Like SAGE III, ACE-FTS does not account for the $NO_2$
diurnal variability along the line of sight, and these two versions have a relatively uniform difference for all altitudes. The diurnal effect error is similar to what Brohede et al. (2007) found when comparing SAGE II and III to OSIRIS. In a recent study by Dubé et al. (2021), they attempted to correct for this effect in SAGE III/ISS $NO_2$ measurements, which improved the agreement between SAGE III and OSIRIS below 20 km. However, they also noted that the corrections weren't sufficient to account for all the differences at these altitudes.

The scale factors applied in this comparison were derived using individual months/years of the simulation. We found little difference when using monthly climatological scale factors except for the year 2019 at altitudes between 10 - 20 km in the tropics and Northern Hemisphere (NH) midlatitude, where the difference can reach 2% in the tropics and 7% in the NH (not shown). It is therefore our recommendation that it is sufficient to use the global climatology when correcting for the $NO_2$ diurnal variation in validation studies. However, we recommend using the month/year scale
factor when merging multiple datasets for trend studies as differences caused by the QBO variability can be as large as 7% below 20 km. Scale factors for specific years are also valuable when focusing on a specific month and region.

### 4.2 Diurnal Scaling Factors for $O_3$

This section presents the diurnal scaling factors for $O_3$, including their temporal and spatial variability. We illustrate the importance of the diurnal correction for $O_3$ in Figure 8, which shows the difference between the simulated $O_3$ at
sunrise and sunset and the simulated ozone at 1:30 in the afternoon, which is the approximate time of the MLS daytime overpass, and 2:30 am, corresponding to the MLS night-time overpass. This difference represents the expected impact of the diurnal variability when comparing SAGE III/ISS observations with MLS daytime observations. Below approximately 25 km, the differences within latitude bands are small compared to the variability within the bands shown by the error bars. However, the average differences can also exceed 2 percent below 25 km in the tropics. The
differences compared to MLS daytime observations increase above 25 km, although they remain within +/- 10% until approximately 60 km (Fig. 8a-c). The sign of the difference switches with altitude. The sunrise $O_3$ falls within a few percent of the MLS nighttime values for altitudes below 50 km, while somewhat larger relative differences are present for the sunset $O_3$ between 35 and 50 km (Fig. 8d-f).

Figure 9 shows the shape of the sunrise diurnal scale factors for $O_3$ at 35 km. The shape of the factors at the equator is similar for January and July. Values dip shortly after sunrise (SZA=90°), rise over the course of the day to an afternoon peak, and then decrease until sunset. There is relatively little change in the nighttime (abs(SZA) > 90°).





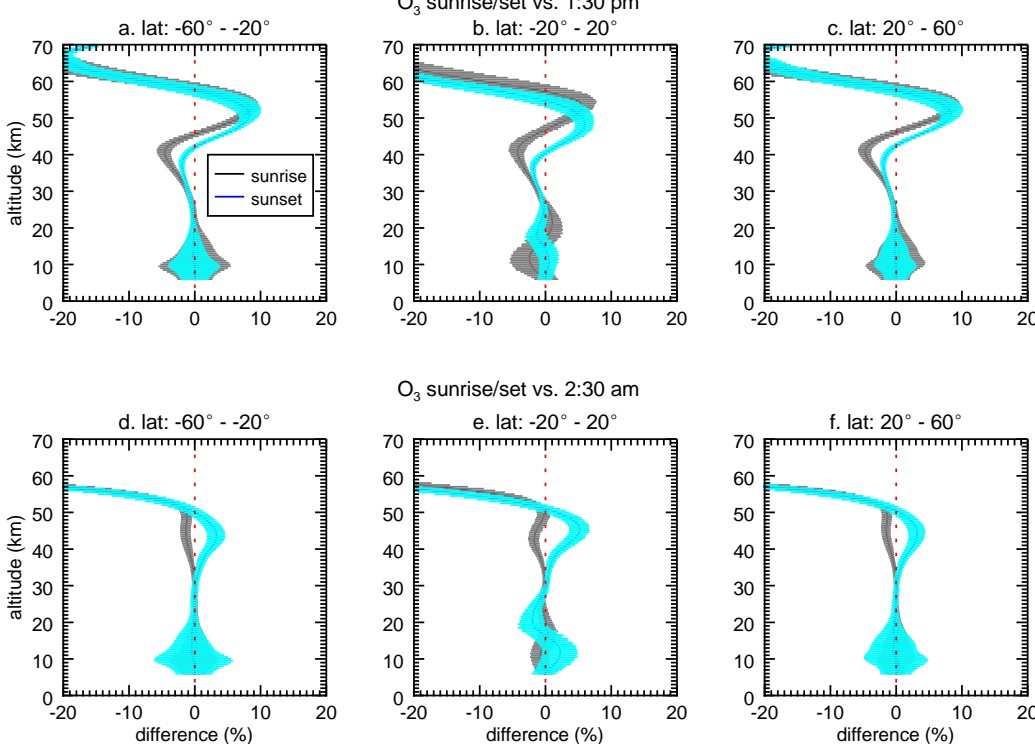

**Fig. 8: The simulated percent difference in O₃ between sunrise (black) or sunset (blue) versus (a-c) 1:30 pm or (d-f) 2:30 am for three latitude bands for all months of 2019. Error bars represent the variability within the band.**


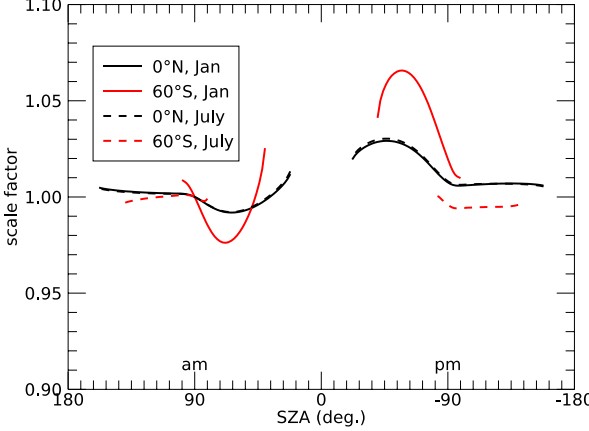

**Fig. 9: Sunrise scale factors for O₃ at 35 km as a function of SZA for January (solid lines) and July (dashed lines) at the equator (black) and 60°S (red).**





This shape is even more pronounced at 60°S in January. The stronger variability at 60°S in Southern Hemisphere summer is consistent with the results of Schanz et al. (2014). The daytime increase to an afternoon maximum is consistent with the results of Haefele et al. (2008) and Parrish et al. (2014). Haefele et al. (2008) point out that production of odd oxygen by photolysis can explain this increase, since $O_x$ is primarily $O_3$ at this altitude. The dip after sunrise is consistent with the findings of Pallister and Tuck (1983), who attribute it to the photodissociation of

$NO_2$, followed by reaction of $O_3$ with NO. The interannual variability in the $O_3$ diurnal cycle diminishes below approximately 50 km (Fig. S5).

### 4.3 Application of $O_3$ Diurnal Scale Factors

To illustrate the utility of the derived $O_3$ scaling factors, we compare SAGE III and MLS at different times with and without the diurnal corrections. The coincidence criteria used for all comparisons shown here is similar to those

described in Sect 4.1.2. MLS profiles were converted to number density and geometric altitude using MLS geopotential altitude, pressure, and temperature profiles. Figure 10 (top row) shows a comparison between SAGE III/ISS $O_3$ observations at sunrise and sunset with daytime and night-time MLS observations with no corrections for the diurnal cycle applied. The comparisons between the different time of day pairs diverge above approximately 35 km, and exceed 10% for the comparisons to MLS daytime observations above approximately 50 km. In addition, the

sign of the difference between SAGE III/ISS observations and MLS observations is positive above 50 km, although the switch to positive occurs a few kilometers higher for the sunrise SAGE III/ISS vs. night-time MLS case. The bottom row of Fig. 10 shows the same comparison but with the diurnal scaling factors applied to account for differences due to the diurnal cycle. The spread between the different time-of-day pairings is greatly reduced above 35 km, providing a more consistent picture of the SAGE III/ISS versus MLS $O_3$ differences. In general, the difference

between SAGE III/ISS and MLS is less than 5% between 20-45 km. Application of the diurnal scaling factors reveals a consistent high bias in the SAGE III/ISS observations compared to MLS above 50 km.

Wang et al. (2020) reported a larger than expected diurnal magnitude of 5-8% difference between SAGE III/ISS sunset and sunrise measurements in the upper stratosphere that they could not explain. We evaluate the differences in SAGE

III/ISS sunrise versus sunset measurements by comparing how they differ from MLS, similar to Wang et al. (2020), who also used MLS observations as a transfer standard. Figure 11 shows the difference between SAGE III sunset and sunrise $O_3$ observations using MLS daytime (blue) and nighttime (red) observations before and after applying the scale factors. The figure shows a 5-7% difference at altitudes between 40-50 km, similar to the sunrise/sunset differences shown in Figure 7 by Wang et al. (2020). However, the difference is reduced significantly to less than 2%

through most of the 40-50 km range when applying the scale factors. Sunrise/sunset differences are almost indistinguishable when using MLS daytime or nighttime measurements.

We also compared SAGE III to various satellite observations. Fig. 12 shows the percent difference between SAGE III and MLS (night), OMPS-LP, OSIRIS, and ACE-FTS before (top) and after (bottom) applying the diurnal scale factor


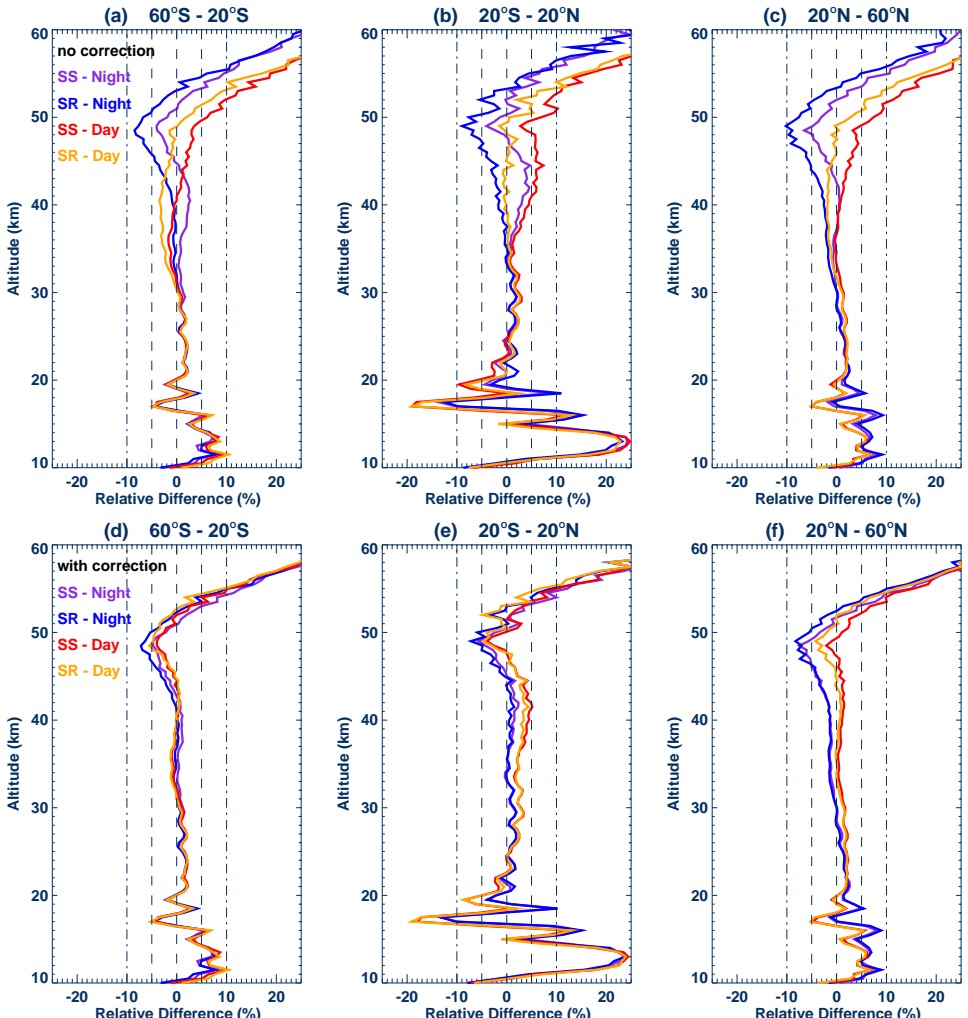

**Fig. 10: (top) Comparison of SAGE III/ISS sunrise (red) and sunset (yellow) O₃ observations with MLS daytime observations. Sunset and sunrise SAGE III/ISS observations are compared to MLS night time observations (purple and blue lines, respectively) at three different latitude zones. Relative difference is SAGE III – MLS and shown in percent. No diurnal corrections are applied in this comparison. (bottom) Same as top row but with the diurnal scaling factors applied.**


corrections. OMPS-LP and OSIRIS are limb scattering instruments that measure the ozone profiles at different times during the day. The figure shows that the difference between SAGE and correlative measurements is mostly within 5% between 20-40 km with some exceptions. ACE-FTS has a larger bias above 45 km similar to Sheese et al. (2017) and Wang et al. (2020), while OMPS LP has over a 10% positive bias between 25-30 km in NH, similar to Wang et al. (2020) and Kramarova et al. (2018). Around 50 km, the differences increase to 10% between SAGE III and OMPS




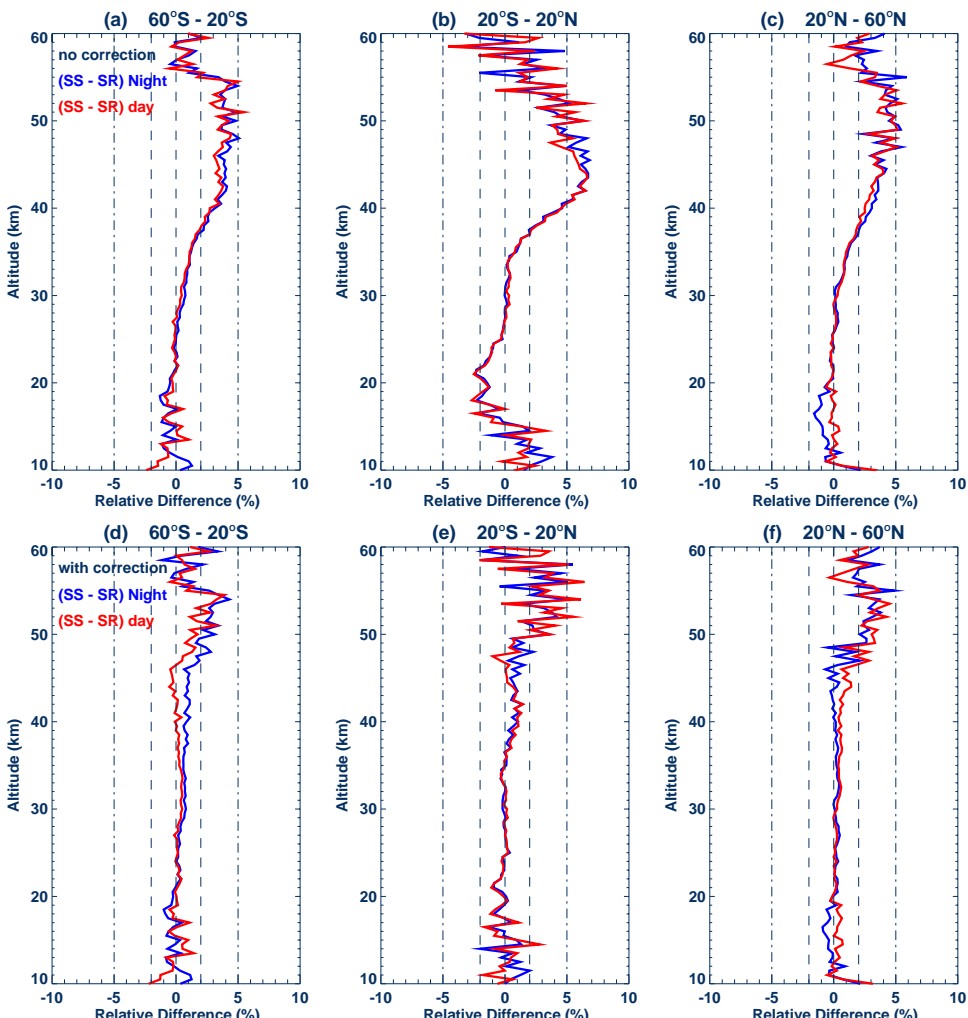

**Fig. 11: (top)** The difference (%) between SAGE III/ISS sunset (SS) and sunrise (SR) O₃ observations in three different latitude zones when MLS daytime (red) and nighttime (blue) observations are used as a transfer standard. No diurnal corrections are applied in this comparison. **(bottom)** Same as top row but with the diurnal scaling factors applied.

LP and ACE-FTS, but the bias compared to OMPS LP is positive at 50 km while the bias compared to ACE-FTS, OSIRIS, and MLS is negative (Fig. 12 top). This difference compared to OMPS LP is largely reduced to within 5% above 35 km once the scale factors are applied (Fig. 12 bottom). This is consistent with the finding of Frith et al (2020) that accounting for the diurnal cycle reduced the differences between SAGE III/ISS and OMPS LP observations. This comparison illustrates the importance of accounting for the diurnal cycle of O₃ when comparing observations from

different times of the day or when merging multiple instruments used for trend studies. Above 50 km, the SAGE III/ISS observations are biased high compared to ACE-FTS and OSIRIS as well as MLS, consistent with the results



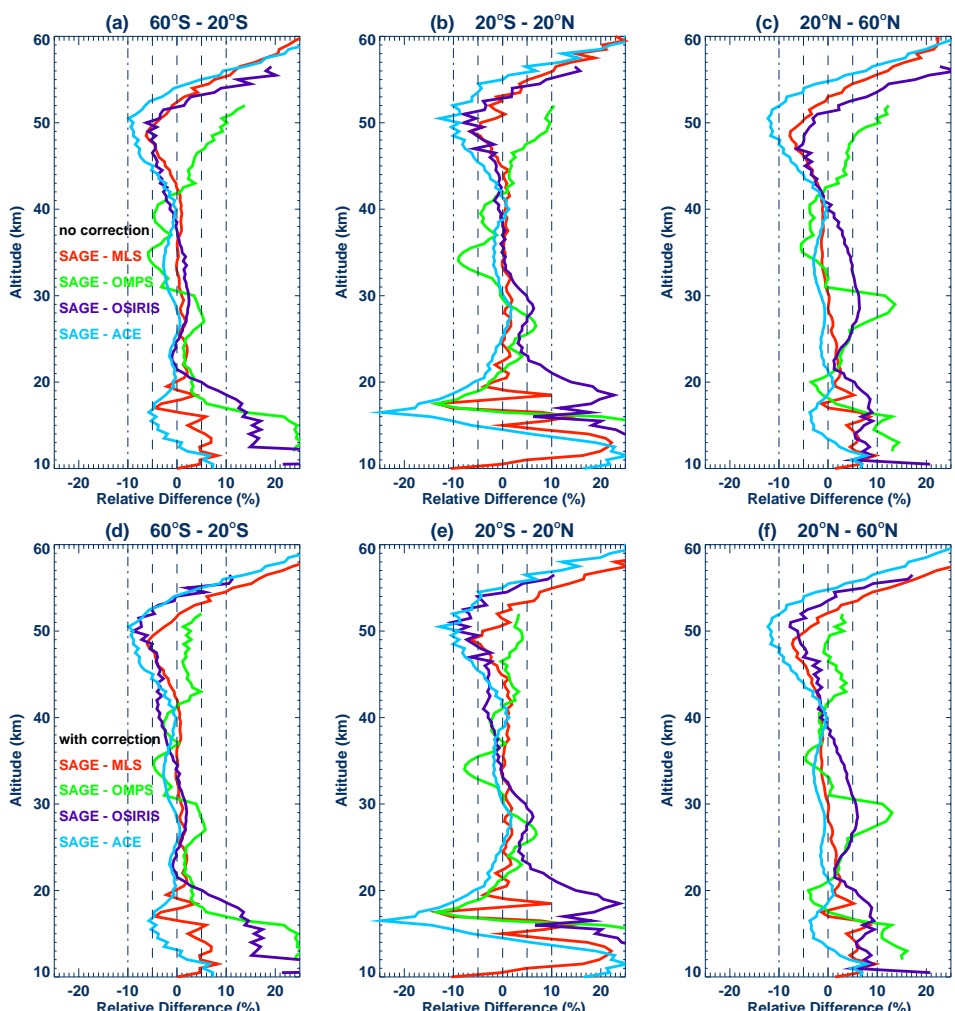

**Fig. 12: (top) Comparison of SAGE III/ISS O₃ observations with MLS night-time observations (red), OMPS LP (green), OSIRIS (violet), and ACE-FTS (blue) at three different latitudinal zones. Relative difference is SAGE – instrument and shown in percent. No diurnal corrections are applied in this comparison. (bottom) Same as top row but with the diurnal scaling factors applied.**


in Fig. 10. As shown in Fig. S5, the variability of the scale factors is very small below 50 km. It is therefore our recommendation that using global climatology is sufficient to accurately correct for the ozone diurnal variations.

## 5 Summary and Conclusions

We used the GEOS-GMI global atmospheric chemistry model simulation to develop diurnal scale factors for 2017-2020 to account for differences between SAGE III/ISS and other observations due to the diurnal cycles of $NO_2$ and $O_3$. These scale factors provide a straightforward method for comparing observations from different times of day as



they provide the ratios of $O_3$ and $NO_2$ at each solar zenith angle to their values at sunrise and sunset based on the simulated diurnal variability, and account for dynamically and chemically driven variability. Furthermore, merging

of the SAGE-measured photochemically active species, such as $NO_2$ and $O_3$ (above 45 km) with other satellite measurements is inherently difficult because of their strong diurnal variations. The diurnal scale factors can be used to scale all measurements to the same time of the day. We validate the model simulation against SAGE III/ISS v5.2 retrievals and other observations, and find good overall agreement in the profile shapes of $NO_2$ and $O_3$.

The scale factors vary with altitude, latitude, and month, and are available for individual years to account for interannual variability. We also provide a monthly climatology based on the 2017-2020 average, which can be used to compare observations outside the 2017-2020 range. Interannual variability in the diurnal cycle of $NO_2$ in the lower stratosphere is linked to the QBO. Overall, however, the interannual variability in the diurnal scale factors is relatively small in the stratosphere, especially for $O_3$, so climatological scale factors are likely sufficient for most applications.

However, accounting for IAV might be necessary when merging different $NO_2$ datasets that are used for trend studies at altitudes above 40 km.

We show that application of the diurnal scale factors for $NO_2$ improves this agreement between SAGE III/ISS and OSIRIS $NO_2$ observations, and the consistency between the comparisons for sunrise and sunset observations. The

comparison between SAGE III/ISS and MLS ozone shows large differences in the magnitude and sign of the disagreement depending on whether sunrise or sunset SAGE III/ISS observations and daytime or night-time MLS observations are considered. Applications of the diurnal scale factors removes much of this variability, providing a more consistent view of the SAGE III/ISS versus MLS $O_3$ differences. Diurnal corrections can also account for the significant and unexplained differences in SAGE III/ISS sunrise versus sunset ozone measurements reported by Wang

et al. (2020). The scaling factors used in this study are now available as a tool to facilitate comparison between observations from different times of day. SAGE III/ISS V5.2 ozone agrees well with correlative measurements, with differences well within 5% between 20-50 km when corrected for diurnal variability. Similarly, the SAGE III/ISS V5.2 $NO_2$ agreement with correlative measurements is mostly within 10%. The larger difference between SAGE III and OSIRIS below 25 km is caused by the diurnal effect from the variation of the SZA, and hence the $NO_2$, along the

line of sight, which is neglected in the SAGE III retrieval and requires further corrections (Dubé et al., 2021).

### Data Availability

The diurnal scale factors described in this work are available at https://avdc.gsfc.nasa.gov/pub/data/project/GMI_SF/. SAGE III/ISS data is available from https://asdc.larc.nasa.gov/project/SAGE%20III-ISS. OSIRIS data is available from      https://research-groups.usask.ca/osiris/data-products.php.      OMPS-LP      data      is      available      from

https://disc.gsfc.nasa.gov/datasets/OMPS_NPP_LP_L2_O3_DAILY_2/summary. ACE-FTS data is available from http://www.ace.uwaterloo.ca/data.php. MLS data is available from https://mls.jpl.nasa.gov/eos-aura-mls/data-access.



**Author Contributions**

SAS, GT, LDO and MS designed the study. SAS created the scaling factors. SAS and GT performed the analyses
and LDO performed the model simulation. RS provided the ozonesonde comparison. RD and DF contributed
scientific discussion of the SAGE III/ISS observations and CES contributed scientific discussion of the OSIRIS
observations. SAS and GT wrote the manuscript and all co-authors contributed comments and editing of the
manuscript.

**Competing Interests**

The authors declare that they have no conflict of interest.

**Acknowledgements**

This work was supported by NASA grant 80NSSC18K0711. The ACE mission is supported by the Canadian Space
Agency. GEOS-GMI development is supported by the NASA Modeling, Analysis, and Prediction (MAP) program
and computing resources were provided by the NASA Center for Climate Simulation (NCCS). We thank the
instrument teams that provided the SAGE III/ISS, MLS, OSIRIS, ACE-FTS, and OMPS LP data.

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
