# Peer review of "SAGE III/ISS Ozone and NO2 Validation Using Diurnal Scaling Factors"

_Atmospheric Measurement Techniques, 2022_

## Referee Comment (RC1)

Strode et al. developed in their study a global set of $NO_2$ and $O_3$ diurnal scaling factors accounting for the diurnal variability of $NO_2$ and $O_3$ concentrations in the atmosphere. The scaling factors were generated by using a 4D global atmospheric chemistry model, and are publicly available in dependence of solar zenith angle, latitude, and altitude. This work is relevant since the authors close a gap, which up to now do not allow an accurate comparison of different measurements (satellite vs satellite or satellite vs ground-based) of $NO_2$ and $O_3$ taken at different times of the day. The authors show, that the utilization of these scaling factors for comparisons (SAGE III/ISS, OSIRIS; MLS, OMPS and ACE-FTS) tremendously reduce the difference between the compared $NO_2$ and $O_3$ concentrations. Furthermore, Strode et al. could show that the interannual variability of $NO_2$ scaling factors is very likely to be correlated to the quasi-biennial oscillation (QBO). I recommend this paper to be published in Atmospheric Measurement Techniques, after the following minor points of criticism will have been addressed.

General remarks:

1) Consistently use $NO_2$ and $O_3$ OR nitrogen oxide and ozone. I would recommend to firstly mention nitrogen oxide ($NO_2$) and ozone ($O_3$) and the switch to only $NO_2$ and $O_3$.
2) For a better readability use consistent order of discussed trace gas in section 1, section 2.1.1, section 2.3 and figure 1. FIRST $NO_2$ and SECOND $O_3$.
3) Change the order of instruments in section 2.1 to be consistent with the order of mention later in the manuscript: SAGE, OSIRIS, ACE-FTS, MLS, OMPS LP
4) Please revise the reference section regarding missing doi or page numbers.

Specific remarks:

1) Title: Capitalize "Using".
2) Line 41 – 54: Please mention the order of magnitude for both the $NO_2$ and $O_3$ diurnal variation/photochemistry to get an idea of the difference of both species.
3) Section 2: Underline difference between experimental data collection and simulation by using 2 sub-sections "2.1 Instruments and observation" and "2.2 Simulation and scaling factors" instead of 2.1-2.3. 2.2 can then be split up into "GEOS Model Simulation" and "Scaling Factor Calculation".
4) Section 2.1: Better indicate whether the description regards $O_3$, $NO_2$ or both, especially for the used retrievals.
5) Line 151 – 169: Is the dynamical tendency of $NO_2$ neglected in the analysis due to the dominance of the chemistry? This is not clear here.
6) Line 188 – 191: For me the method is not clear here. Do you just take the best fitting data to compare model and observation and not the SZA=90° data? Is this admissible in this context? The "real" 90° value is unknown, isn't it? Please clarify.
7) Line 221 – 230: Shift complete paragraph into the introduction or shorten it.
8) Line 237: "the $O_3$ peak" instead of "the peak $O_3$".
9) Line 248: Change the section title. The result part already starts in section 3 with the model validation. Maybe "Data evaluation".
10) Section 4.1.2: define the parameter "sunrise scale diff" as used in the figures.
11) Line 319: Change "SAGE III/ISS sunrise (SR) and sunset (SS) $NO_2$ and OSIRIS and ACE320 FTS observations" to "SAGE III/ISS sunrise (SR) and sunset (SS) and OSIRIS and ACE320 FTS $NO_2$ observations".
12) Line 365 – 381: It would be helpful to note the difference in magnitude of the scale factors when comparing $O_3$ and $NO_2$.

13) Figure 8: Colors of legend and data are not the same.
14) Figure 9: Suggest to use a wider y-axis-range.
15) Figure S4: Mention that the shown data is $NO_2$ data.
16) Figure S5: SZA=60 → SZA=60° (unit missing)

---

## Author Response (AR1)

Response to Comments on "SAGE III/ISS Ozone and NO$_2$ Validation Using Diurnal Scaling Factors"

We thank the reviewers for the valuable comments.  Our responses to each point are provided below.  The reviewer comments are in gray, and our responses are in bold black text.

Response to Reviewer 1 Comments

Strode et al. developed in their study a global set of NO$_2$ and O$_3$ diurnal scaling factors accounting for the diurnal variability of NO$_2$ and O$_3$ concentrations in the atmosphere. The scaling factors were generated by using a 4D global atmospheric chemistry model, and are publicly available in dependence of solar zenith angle, latitude, and altitude. This work is relevant since the authors close a gap, which up to now do not allow an accurate comparison of different measurements (satellite vs satellite or satellite vs ground-based) of NO$_2$ and O$_3$ taken at different times of the day. The authors show, that the utilization of these scaling factors for comparisons (SAGE III/ISS, OSIRIS; MLS, OMPS and ACE-FTS) tremendously reduce the difference between the compared NO$_2$ and O$_3$ concentrations. Furthermore, Strode et al. could show that the interannual variability of NO$_2$ scaling factors is very likely to be correlated to the quasi-biennial oscillation (QBO). I recommend this paper to be published in Atmospheric Measurement Techniques, after the following minor points of criticism will have been addressed.

**We thank the Referee for the thoughtful comments and respond to individual points below.**

General remarks:

- Consistently use NO$_2$ and O$_3$ OR nitrogen oxide and ozone. I would recommend to firstly mention nitrogen oxide (NO$_2$) and ozone (O$_3$) and the switch to only NO$_2$ and O$_3$.

**We now use NO$_2$ and O$_3$ throughout the text except in the initial mention and in instrument names.**

- For a better readability use consistent order of discussed trace gas in section 1, section 2.1.1, section 2.3 and figure 1. FIRST NO$_2$ and SECOND O$_3$.

**We re-organized the text in the second and third paragraphs of Section 1 to discuss NO$_2$ and the O$_3$. Specifically, we moved the sentence about previous studies using PRATMO to an earlier paragraph.  We reworded and moved the sentence about NO$_2$ bias before the ozone bias discussion in Section 2.1.1. This sentence now states: "Dubé et al. (2021) reported that the SAGE III/ISS NO2 V5.1 is over 20% biased high in much of the mid-stratosphere even when accounting for diurnal variability".**

**We retain the order of Section 2.3 (now 2.2.2) and figure 1 because the larger relative impact of dynamics for the O$_3$ diurnal cycle is important for motivating this discussion.**

- Change the order of instruments in section 2.1 to be consistent with the order of mention later in the manuscript: SAGE, OSIRIS, ACE-FTS, MLS, OMPS LP

**We now use the suggested order.**

- Please revise the reference section regarding missing doi or page numbers.

**We added the doi links.**

Specific remarks:

- Title: Capitalize "Using".

**Done**

- Line 41 – 54: Please mention the order of magnitude for both the $NO_2$ and $O_3$ diurnal variation/photochemistry to get an idea of the difference of both species.

**We add the following to the $NO_2$ discussion: "Using PRATMO, Dubé et al. (2021) showed a diurnal range exceeding a factor of 3 for $NO_2$ at the equator at 30 km." and we add to the $O_3$ discussion: "Frith et al (2020) found the $O_3$ diurnal cycle exceeds 15% in the upper stratosphere near the edge of the polar day."**

- Section 2: Underline difference between experimental data collection and simulation by using 2 sub-sections "2.1 Instruments and observation" and "2.2 Simulation and scaling factors" instead of 2.1-2.3. 2.2 can then be split up into "GEOS Model Simulation" and "Scaling Factor Calculation".

**We combined sections 2.2 and 2.3 into subsections of "2.2 Simulation and scaling factors" as suggested.**

- Section 2.1: Better indicate whether the description regards $O_3$, $NO_2$ or both, especially for the used retrievals.

**We already state "$NO_2$" or "$O_3$" when discussing specie-specific retrievals.**

- Line 151 – 169: Is the dynamical tendency of $NO_2$ neglected in the analysis due to the dominance of the chemistry? This is not clear here.

**We now clarify: "Our scaling factors for $NO_2$ also include both chemical and dynamical effects, but for $NO_2$, the chemical tendency is dominant throughout the profile (Fig. 1d-f)."**

- Line 188 – 191: For me the method is not clear here. Do you just take the best fitting data to compare model and observation and not the SZA=90° data? Is this admissible in this context? The "real" 90° value is unknown, isn't it? Please clarify.

**We take the data from the grid box at the observation latitude within 8 grid boxes longitudinally of the observed longitude whose SZA best matches that of the occultation measurements (90°). Thus we are selecting the best fitting SZA, not the best fitting data. We think this is reasonable since we are using the model to define scaling factors relative to SZA, so it is important to validate the model for the observed SZA. To clarify this, we now state "then finding the grid box whose SZA best matches the SAGE III/ISS SZA (±90°)..."**

- Line 221 – 230: Shift complete paragraph into the introduction or shorten it.

**We prefer to keep this paragraph in place since it provides the background for the results presented in this section.**

- Line 237: "the $O_3$ peak" instead of "the peak $O_3$".

**We made this change.**

- Line 248: Change the section title. The result part already starts in section 3 with the model validation. Maybe "Data evaluation"

**We prefer to keep the section title as "Results" since this section includes the main results of our study.**

- Section 4.1.2: define the parameter "sunrise scale diff" as used in the figures.

**We now clarify in the caption of Fig. 6b: "percent difference from climatology in the sunrise scaling factors (denoted "sunrise scale diff" in the axis labels)"**

- Line 319: Change "SAGE III/ISS sunrise (SR) and sunset (SS) $NO_2$ and OSIRIS and ACE320 FTS observations" to "SAGE III/ISS sunrise (SR) and sunset (SS) and OSIRIS and ACE320 FTS $NO_2$ observations".

**Changed**

- Line 365 – 381: It would be helpful to note the difference in magnitude of the scale factors when comparing $O_3$ and $NO_2$.

**We added: "We note that the y-axis range of Fig. 9 covers a smaller range of values than that of Fig. 4, which showed $NO_2$ scale factors."**

• Figure 8: Colors of legend and data are not the same.

**We modified the figure so that the mean values, whose colors match the legend, are visible above the errorbars.**

• Figure 9: Suggest to use a wider y-axis-range.

**We selected this axis range in order to show sufficient detail in the figure. As noted above, we now point out that this y-axis range is smaller than that of Fig. 4.**

• Figure S4: Mention that the shown data is $NO_2$

**We added this information to the caption.**

• Figure S5: SZA=60 à SZA=60° (unit missing)

**We added the degree sign.**

Response to Reviewer 2 Comments

This is a very well written paper, and is a good fit for AMT. It demonstrates the utility of global chemistry-climate models to scale measurements of diurnally varying species for comparing or merging data sets. I would recommend the paper for publication after just a few minor details are addressed, as given below.

**We thank the referee for the helpful comments and respond to individual points below.**

Line 18: What do you mean by "variability in the shape"? I think you can leave out "in the shape"

**We removed "shape".**

Lines 48-49: Please be more specific about the findings

**We elaborated this sentence by adding: "due to changes over time in the relative frequency of sunrise and sunset measurements combined with diurnal variability"**

Line 107: please give approximate altitudes

**We added: (~18-59 km).**

Section 2.1.4: Why is v3.6 of the ACEFTS data being used instead of the more current v4.2? Version 3.6 O3 exhibits a drift in the upper stratosphere, and there were biases between the two versions (https://doi.org/10.5194/amt-15-1233-2022)

**When we performed our analysis, Version 3.6 was the recommended version for validation studies [Wang et al., 2020]. In addition, the positive bias for ozone in the middle stratosphere is approximately 3% in version 3.6 but 2-9% in version 4.1 [Sheese et al., 2022, https://doi.org/10.5194/amt-15-1233-2022]. We added this explanation to the description of ACE-FTS in section 2.1.3.:**

**"We used Version 3.6 instead of V4.1 since it was the recommended version for validation studies [Wang et al., 2020]. In addition, the positive bias for ozone in the middle stratosphere is approximately 3% in version 3.6 but 2-9% in version 4.1 [Sheese et al., 2022]."**

Lines 133-135: does this mean that this is like a specified dynamics run?

**There are multiple methods for constraining or specifying the dynamics in a global atmospheric chemistry model. The specific method for constraining the meteorological fields in our simulation is described in the Orbe et al [2017] reference.**

Line 160: Undoubtedly, it would be important above 50 km as well. Should say something like "within the SAGE observation range it's only important between 40 and 50 km."

**We added "within the SAGE III/ISS observation range"**

Figure 1: are these at a specific sza or averaged over all szas?

**This is the amplitude of the diurnal cycle, calculated as the maximum of the monthly mean diurnal cycle minus the minimum of the monthly mean diurnal cycle at each grid box, and then averaged over longitudes and latitude bands. Multiple SZAs are thus considered since the maximum and minimum occur at different SZAs. We updated the text describing the figure in section 2.2.2 to clarify that we are using the monthly mean diurnal cycle:**

**"Figure 1 compares the amplitude of the diurnal cycle, defined here as the maximum of the monthly mean diurnal cycle minus the minimum"**

Line 197: please be more specific about the findings

**We added "and that accounting for diurnal variability along the line of sight can reduce the bias below 30 km by over 10%"**

Lines 221-230: please be more quantitative about previous findings

**We added additional information to this discussion including percent differences. We now state: "Parrish et al. (2014) found reasonable agreement between the simulated $O_3$ diurnal cycle at Mauna Loa, Hawaii with microwave ozone profiling radiometer (MWR) observations at most levels, with most of the modelled and measured values agreeing to within 1.5% of the midnight value. However, between 39 and 43 km, the morning versus night differences in**

**the MWR observations are 2-3% higher than in the model." In the discussion of the Frith et al [2020] results, we now state: "They found good overall agreement with the structure of the MLS differences, generally within 2%, while the simulated sunrise/sunset ratio differed from that of SAGE III/ISS above approximately 2 hPa but agreed within approximately a percent below 2 hPa."**

Section 3.2 in general: please be more quantitative in your descriptions

**In addition to the changes made for the previous comment, we also add additional quantitative information to the description of our model comparison to the SAGE III/ISS observations in this section. We added the following:**

**"The largest percent difference in this range for the sunrise observations is 13% and occurs at 20 km for the 20°S-20°N band. The largest percent difference in this range for the sunset observations is 12% and occurs at 20.5 km for the 20°S-20°N band."**

**"The SAGE III/ISS sunrise and sunset averages for this latitude band reaches a peak of $4.5*10^{12}$ molec cm$^{-3}$ at 26.5 km while the model reaches a peak of $4.3*10^{12}$ molec cm$^{-3}$ at 26 km."**

**"For June-Aug., the model agrees with the observations within 30% between 20 and 50 km, with the largest percent difference occurring at 20 km."**

Section 4.1.1: Why was 35 km chosen to be shown? Seems like it might be more interesting to see closer to 40-45 km where amplitudes are largest.

**We chose 35 km because it is more relevant to the comparison with OSIRIS observations.**

Figs 5,6 captions: sza values are missing the degree sign

**We added the degree sign.**

Line 295: It's a touch confusing. Would recommend something like "variability is largest at 60deg S"

**We reworded this sentence to state: "At 60°S, the differences between individual years and climatology reach values above 20% near 10-20 km (Fig. 6d)."**

Fig 7: These plots could very easily be made much more intuitive to read. Please color coordinate/use different line styles to group the plots. Like all OSIRIS comparisons could be one color but different line styles for different criteria, and the ACE could be a different color with the same line styles.

**We modified the lines in Fig. 7 for SAGE III – OSIRIS so that the "no correction" and "with correction" sunrise comparisons are both red and the "no correction" and "with correction" sunset comparisons are both blue, but we use solid lines for the comparisons with no correction and dashed lines for the comparisons with correction. Consequently, it is now easier to see the effect of the diurnal corrections by comparing solid versus dashed lines of the same color.**

Line 361: The sentence "The sign of the difference switches with altitude." should either be elaborated on or deleted.

**We reworded this sentence to say: "The sign of the difference switches between positive and negative depending on altitude."**

Line 367: abs(SZA) > 90$^{\circ}$ could be |SZA| > 90°

**We changed abs(SZA) to |SZA|**